# Preliminary steps of the development of a Minimum Uniform Dataset applicable to the international wheelchair sector

María Luisa Toro-Hernández[1◉], Nancy Augustine[1,2◉], Padmaja Kankipati[3,4], Patricia Karg[1,2], Karen Rispin[4,5¤], Richard M. Schein[2,4], Krithika Kandavel[1,2], Megan E. D'Innocenzo[1,2], Mary Goldberg[1,2], Jonathan Pearlman[1,2]*

1 International Society of Wheelchair Professionals, Pittsburgh, Pennsylvania, United States of America, 2 School of Health and Rehabilitation Science, University of Pittsburgh, Pittsburgh, Pennsylvania, United States of America, 3 Specialized Mobility Operations and Innovation Pvt. Ltd, Bangalore, India, 4 Member of the International Society of Wheelchair Professionals, Pittsburgh, Pennsylvania, United States of America, 5 School of Arts and Science, Letourneau University, Longview, Texas, United States of America

◉ These authors contributed equally to this work.
¤ Current address: Ideas World, Colorado, United States of America
* jpearlman@pitt.edu

**Data Availability Statement:** All relevant data are within the manuscript and its Supporting Information files.

## Abstract

Assistive products outcomes are needed globally to inform policy, practice, and drive investment. The International Society of Wheelchair Professionals developed a Minimum Uniform Dataset (MUD) for wheelchair services worldwide with the intent to gather data that is comparable globally. The MUD was developed with the participation of members from around the globe and its feasibility piloted at 3 sites. Three versions of the MUD are now available—a short form with 29 data points (available in English, Spanish, and French) and a standard version with 38 data points in English. Future work is to validate and complete the translation cycles followed by promoting the use of the MUD globally so that the data can be leveraged to inform policy, practice and direct investments.

## Introduction

The World Health Organization (WHO) defines rehabilitation as a strategy to optimize an individual's functioning [1]. WHO has also recognized that rehabilitation is a prerequisite to achieve health and well-being for all [1]. There is a vast gap of evidence and outcomes in rehabilitation, including rehabilitation needs of the population, cost-benefit analysis, and measures of impact through standardized measures of rehabilitation [1–3]. Absence of reliable data and information poses significant barriers to the development and monitoring of effective rehabilitation policies, programs, and the realization of the right to the highest attainable health [2, 4, 5]. Therefore, reliable data and statistics are needed to achieve the United Nations Sustainable Development Goals (SDGs) and to support the scale up of rehabilitation in health systems, especially in low- and middle-income countries (LMICs) [1, 6]. For many people, access to assistive products (AP), and learning how to use them, is needed to realize the right to fully

**Funding:** This research was made possible by the generous support of the International Society of Wheelchair Professionals (ISWP) and the University of Pittsburgh. University of Pittsburgh scientists are working with the U.S. Agency for International Development (USAID) under sub-awards to develop the International Society of Wheelchair Professionals, a global network to ensure a level of standardization, certification and oversight, to teach and professionalize wheelchair services, and to build affiliations to put better equipment in the right hands. Since 2002, USAID has granted more than $45 million to improve wheelchairs and wheelchair services worldwide. The sub-awards are: Agreement No. APC-GM-0068 and Agreement No. APC-GM-0107, presented by Advancing Partners & Communities, a cooperative agreement funded through USAID under Agreement No. AIDOAA-A-12-00047, beginning Oct. 1, 2012; and FY19-A01-6024, presented through University Research Co. LLC Health Evaluation and Applied Research Development (HEARD) Project. HEARD is funded by United States Agency for International Development (USAID) under cooperative agreement number AID-OAA-A-17-00002. The project team includes prime recipient, University Research Co., LLC (URC) and sub-recipient organizations. The contents of the findings of this study are the sole responsibility of Advancing Partners & Communities, University Research Co., LLC and the University of Pittsburgh and do not necessarily reflect the view of USAID or the United States Government. Jonathan Pearlman (JP) is the recipient of the above-mentioned awards. The funder provided support in the form of salaries for authors NA, JP, MG, KK, MLTH, RS, MD, but did not have any additional role in the study design, data collection and analysis, decision to publish, or preparation of the manuscript. The specific roles of these authors are articulated in the 'author contributions' section. PK participated as a volunteer and not as part of her role with SMOI.

**Competing interests:** NO authors have competing interests. PK affiliation with SMOI (Specialized Mobility Operations and Innovation Pvt. Ltd,) does not alter our adherence to PLOS ONE policies on sharing data and materials.

participate in their communities [1, 7–9]. Access to AP across the life span is key to reducing inequalities within and among nations [10–13]. Context-specific evidence is needed to inform the development of policies and programs that warrant appropriate access to AP [10, 14]. Specific to wheelchairs, there is a need for comparable wheelchair provision data, mostly in LMICs, to strengthen the evidence on different interventions [15, 16].

One strategy to measure rehabilitation outcomes through time is large datasets; that is, multiple rehabilitation service sites using the same outcome measures in order to compare rehabilitation outcomes among type of services (i.e., specialized rehabilitation unit vs. non-specialized), health condition, sex, age, and with the potential to assess the interplay of different determinants [5, 17–21]. These large datasets pose an opportunity to generate practice-based evidence to develop benchmarks for the sector [17, 18]. Most large rehabilitation data sets are in high-resourced settings and collect demographic, hospitalization, diagnostic, functional status data, but do not measure AP use at discharge [17, 18, 20, 22]. The use of AP has been identified globally as an overlooked outcome in people who have sustained a spinal cord injury [23]. An exception is the US National Spinal Cord Injury Database which collects data to study the course of spinal cord injury, health service delivery (including AP), and outcomes [24]. Outcomes comparable among different settings [2, 25, 26] with a special emphasis on LMIC [15] are needed to strengthen the evidence on AP. For instance, there is a need to measure the met and unmet need and to monitor and evaluate initiatives longitudinally, including measuring cost-effectiveness to be able to prioritize and promote investment [10, 14, 27–30]. An example is the Assistive Technology Needs Assessment proposed by the World Health Organization (WHO) to measure at a population base the proportion of people who self-identify a need for AP and current AP product satisfaction [28]. This tool, however, is not meant to be used to measure outcomes at multiple time intervals [28]. There is little research to support current wheelchair service delivery recommendations (e.g. WHO guidelines, professional organizations guidelines), and the evidence that exists commonly involves small sample sizes, inconsistent definitions, and different outcome measurements [16, 31]. Most research has been exploratory, including single subject designs in high-income settings, which provide weak evidence [31–33]. Efforts targeted at LMICs to measure wheelchair provision outcomes longitudinally have used different measurements, and a significant group of people were lost to follow-up, making it impossible to aggregate the data [34–37]. Outcomes that are important for users and families is also lacking [33], and instead, outcomes focus on the interface between the user and the wheeled mobility device [38], the working condition of the wheelchair [39, 40], or outcomes related to specific mobility skills interventions [41]. There are two recent large dataset examples specific to wheelchair service in high-income settings. First, in the US, the Functional Mobility Assessment and Uniform Dataset (FMA/UDS) is a Wheeled Mobility and Seating registry responding to the country's context of wheelchair service provision which has been created for quality assurance and to understand what type of mobility devices promote best health and participation according to health condition and other circumstances [42]. The data collected is based on the Functional Mobility Assessment [22, 43] and a uniform dataset [42]. Data is collected at the time of assessment for a new device and periodically afterwards as a follow-up measure [42]. Second, in the UK, the National Wheelchair Data Collection initiative started in 2015 to improve transparency and benchmarking by collecting, in a centralized manner, expenditure, access, volume, and wheelchair user experience [44]. For LMICs, and based on consensus wheelchair provision guidelines, the World Health Organization proposed general intake and follow-up forms for wheelchair services that may be useful to gather global data uniformly [45–47].

A scoping report on global access to AP stated that "the needs of a comprehensive wheelchair service are not widely understood in any country" [27], which may result in low quality

of rehabilitation services both in quantity and quality [4]. A full picture of the need for appropriate wheelchair products and services, including the policies and personnel that support them, is necessary for strategic planning and capacity building to realize the human right to personal mobility [7, 48]. In fact, a global wheelchair sector stakeholders (e.g., users, WHO, aid agencies, academia, international and local NGOs, humanitarian organizations, professional organizations, service providers, governments) meeting in 2018 held to reflect on past achievements, challenges, current initiatives, and to strategize a future of better access to wheelchairs, identified one of the global priorities to be: "Conduct research and collect data . . . Create a repository of data related to: unmet need; product and service quality; impact of appropriate wheelchair provision on health, quality of life, participation, reintegration into daily living and economic benefit analysis; and promote the use of the data to drive evidence-based practice" [49]. To help bridge the gap in wheelchair provision data gathered and comparable in LMICs, the International Society of Wheelchair Professionals (ISWP) [50] developed a Minimum Uniform Dataset with the objective to promote the global use of a common language to strengthen the evidence [15] and to foster comparison among interventions [16]. The purpose of this manuscript is to present the development process and final composition of ISWP's Minimum Uniform Dataset (MUD).

## Methods and results

The Minimum Uniform Dataset (MUD) was developed through an iterative process by stakeholders composed from ISWP's global membership, including the members of the Evidence-based Practice Working Group Data Collection Subcommittee. The iterative process included initial question development, pilots, revisions based on pilots' results, and launch (Fig 1) [51, 52]. Information on the development process was presented at the RESNA conference in 2018 [53].

### ISWP member survey

On May 5, 2015, ISWP sent an e-mail to 353 ISWP wheelchair sector stakeholders who are ISWP members with a survey (S1 File) link to request input on: data they currently collect in their practices; methods of collecting the information; willingness to share de-identified data to help develop common data fields and to use a standardized data management system; methods for using de-identified, aggregated data; and suggested data to be collected. Forty-one individuals (41) responded (11.6%) representing all stakeholder groups and 18 countries, providing a representative sample for this first step of identifying the types and extend of data collection. De-identified data is available in S1 Dataset.

Among 39 respondents who selected the occupation that best described them, 28% (n = 11) were clinicians; 15% (n = 6) worked for non-government organizations; 15% (n = 6) were researchers; and 10% (n = 4) were manufacturers. Thirty percent (n = 12) were grouped in the Other category, which was comprised of suppliers (n = 3), academicians (n = 3), as well as individuals who reported to be in private practice, product manager, or technician. Respondents represented 18 countries: Albania, Argentina, Brazil, Canada, Colombia, Germany, Hong Kong, India, Japan, Kenya, Mexico, Portugal, South Africa, South Korea, Spain, Sri Lanka, United Kingdom, and United States. The survey was distributed only in English; this may have prevented stakeholders whose first language is not English from responding.

Slightly less than half of respondents—47.4% (n = 18)—indicated they collected data on wheelchair skills and abilities of their clients; 52.6% (n = 20) did not, 3 respondents skipped the question.

**Fig 1. ISWP Minimum Uniform Dataset iterative process.**

Overall, the 13 questions had a response rate of 76% or higher. Two questions were responded by 76% of the respondents, one by 88%, and 10 questions were responded by 90% or higher. Among the methods used and reported by 36 respondents (multiple responses accepted): 77.7% (n = 28) kept records on paper; 72.2% (n = 26) conducted user interviews at time of delivery or follow-up (data collection method not specified); and 58.3% (n = 21) conducted user satisfaction, feedback, or impact surveys.

Respondents were shown a list of 25 data points that could be included in a uniform/minimum dataset and asked to select those which should be included; multiple answers were accepted. Eighty percent or more of respondents (n = 33) indicated 21 measures should be included as part of a uniform minimum dataset (Fig 2). No criteria were set a-priori to delete items based on missing responses.

Survey results were presented to the ISWP Advisory Board and Evidence-based Practice Working Group. In November 2015, the group established a subcommittee devoted to the Minimum Uniform Dataset (MUD).

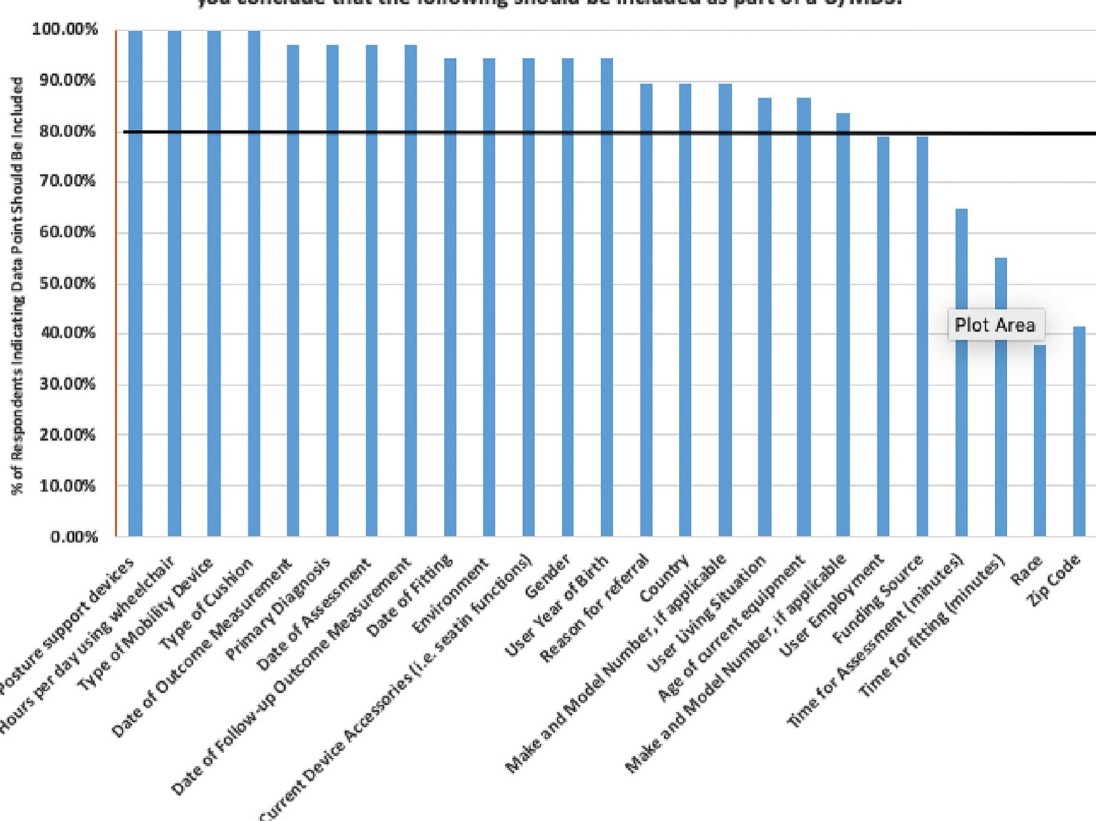

**Fig 2. 2015 ISWP wheelchair sector survey results indicating for each data point the percentage of respondents (n = 33) who thought it should be included in a Uniform Minimum Dataset.**

## Questionnaire development

The ISWP MUD subcommittee, comprised of a cross-section of wheelchair sector stakeholders, used the 21 data points from the 2015 survey as the basis for the first draft of the MUD. The subcommittee also reviewed the WHO wheelchair service provision forms in the WSTP-Basic Wheelchair Service Training Package [45], data fields that World Vision was collecting as part of its ACCESS project [54], WHO Quality of Life [55] and WHO Disability Assessment Schedule 2.0 [56] measures, among other data points they felt service providers should know to inform wheelchair service and provision in international settings. As the number of data points to be collected grew, the subcommittee identified what minimum information should be collected so that a variety of organizations could share the same data points with ISWP.

In late 2015 and early 2016, the subcommittee iteratively refined the questionnaire in preparation for a May 2016 pilot at Bethany Kids Relief and Rehabilitation and Joy Town School for Students with Disabilities in Kenya. Multiple conference calls and response to input resulted in a 21-item questionnaire (Table 1):

**Table 1. ISWP Minimum Uniform Dataset questionnaire data points, 2016.**

| | |
|---|---|
| Year of Birth | Primary mobility aid used* |
| Height | Mobility aid manufacturer name |
| Weight | Mobility aid model/make |
| Gender | Mobility aid serial number |
| Diagnosis* | How long using primary mobility aid* |
| Year of diagnosis | Hours/Day Mobility Aid Used* |
| Highest grade in school completed* | Who Provided Mobility Aid* |
| Current occupational status* | (If wheelchair) Type of Cushion Used* |
| Date of interaction | (If wheelchair) Type of Backrest Used* |
| Purpose of visit* | Settings where mobility aid is used* |
| Physical function ability** | |

*Close-ended question.

**Included yes/no statements for client to answer related to ability to walk 25 feet without support; whether unilateral or bilateral support is needed; whether support is needed only when traveling; and if upper body is impacted and affects ability to self-propel a wheelchair.

## 2016 Kenya pilot

A pilot was conducted in May 2016 by one of the co-authors (KR). The purpose of the pilot was to determine: a) how long it took to administer the questionnaire; b) background of the individual administering it; c) setting/location where it was administered; and c) whether the wording was at a suitable level for comprehension. Students from two schools who used a wheelchair were invited to participate in the pilot. A total of 45 primary school students (average age 11.5 years) and 60 secondary school students (average age 17 years) who all used wheelchairs as their primary mobility device participated. Ethics approval was given by LeTourneau University and LeTourneau's partner organizations in Kenya (LETU IRB: Feb 23, 2017). Participants 18 years or older signed the informed consent and participants younger than 18 provided assent and their guardian signed the written consent. Participants completed paper versions of the 21-item questionnaire. At the primary school, a single data collector administered the ISWP MUD individually to all 45 wheelchair users. An occupational therapist working with the students reviewed the completed documents to identify if there were inaccuracies, specifically with regard to diagnosis. At the secondary school, 60 wheelchair users completed the form in a group session. The research team circled the room to try to clarify questions while students completed the MUD. Those administering the questionnaire recorded comments on an Excel spreadsheet about difficulties with specific questions (e.g., very few respondents knew their height and weight or the year of diagnosis; most countries use meters to measure distance, not feet). S2 Dataset presents the aggregated feedback provided by the data collector.

The 2016 pilot feedback and additional working group and subcommittee review and input resulted in a standard version of the MUD with 40 data points and a short form with 28 data points (Table 2). The subcommittee decided to offer a standard and short form version to provide flexibility while promoting consistency in the implementation as time constraints in services may vary across LMICs. The standard version includes additional questions about wheelchair and cushion manufacturer, make, and model; training received; assistance using a wheelchair indoors and outside; distance traveled in the wheelchair; whether the client takes public transportation; and which transportation methods are used. ISWP also created Excel

**Table 2. ISWP MUD questionnaires, 2017 (bold X indicates data point in standard version only).**

| Data Point | Short Form | Standard Version |
|---|:---:|:---:|
| *Client and wheelchair clinic information* | | |
| Client ID | X | X |
| Client town | X | X |
| Client country | X | X |
| Service provider name | X | X |
| Service provider location | X | X |
| *Date completed* | X | X |
| *Primary purpose of visit** | X | X |
| *Demographics* | | |
| Age/approximate age | X | X |
| Gender* | X | X |
| Education* | X | X |
| Employment* | X | X |
| Living situation* | X | X |
| *Reasons for assistance* | | |
| How long needed mobility aid | X | X |
| Why mobility aid is needed* | X | X |
| Year diagnosis received* | X | X |
| *Mobility aids used* (list of 30 mobility aids grouped by category: Manual wheelchair; electrically powered wheelchair; crank system [cycle]; walking products; braces and artificial limbs; and other mobility aids. Presented as one question with five parts.) | | |
| Rank top 4 currently used most often | X | X |
| Where used (indoors/outdoors) | X | X |
| Use more than one year | X | X |
| Number of hours/days used (grouped) | X | X |
| Number of days/week used (grouped) | X | X |
| *Mobility aid details* | | |
| Manufacturer name, model/make and serial number for each aid used | | **X** |
| *Difficulty walking long distance in past 30 days** | X | X |
| *Assistance* | | |
| Help indoors | | **X** |
| Help outside | | **X** |
| *Questions for current wheelchair users* | | |
| Degree of difficulty pushing* | X | X |
| How client pushes wheelchair* | X | X |
| If not pushing with arms or legs, reasons why* | X | X |
| Places where wheelchair is used currently* | X | X |
| How person received wheelchair* | X | X |
| Agreement statements regarding wheelchair (7) | X | X |
| Overall level of satisfaction with wheelchair | X | X |
| Distance traveled in wheelchair each day | | **X** |
| Whether public/private transportation is used | | **X** |
| If public/private transportation used, what kind | | **X** |
| Whether client has ever fallen out of wheelchair | | **X** |
| Cushion type*, manufacturer, make/model name or number | | **X** |
| Training received about how to use wheelchair | | **X** |
| *Whether survey was completed by client or someone else* | | **X** |

*(Continued)*

**Table 2.** (Continued)

| Data Point | Short Form | Standard Version |
|---|---|---|
| *Name of referral, if applicable* | | X |
| *Cushion details* | | |
| Manufacturer name, model/make for cushion | | X |

*Close-ended question.

workbooks for each version which included drop-down boxes for questions with close-ended responses to facilitate data entry.

## 2017 Indonesia and Kenya pilots

A second pilot of the short form questionnaire was conducted by LeTourneau University in Kenya in May 2017 with 31 primary school students and 64 secondary school students who all use wheelchair as their primary means of mobility. Ethical approval and recruitment and consent processes were as described above. A volunteer used the Excel workbook to administer the questionnaire with the primary students. Secondary school students completed the paper questionnaire with help from volunteers who clarified questions, but did not help with responses, and noted when students said they had difficulty completing a question. Secondary school students took, on average, 11 minutes to complete the questionnaire and had difficulty answering 7 of the 28 data points. S2 Dataset presents the aggregated feedback provided by the data collector.

In February 2017, UCP Wheels for Humanity partnered with the Comprehensive Initiative on Technology Evaluation at the Massachusetts Institute of Technology (CITE-MIT) and Puspadi Bali Foundation to pilot a standard version of the survey (40 data points) with 150 respondents in Bali, Indonesia as part of the Wheelchair User's Voice Project [57]. Ethics approval was given by the University of Pittsburgh (ID: 679—CITE Evaluation of Wheelchair Usage and Rider Experience in Indonesia). The questionnaire was forward translated and administered in Bahasa, and written consent was obtained from all research participants (S2 File presents the MUD in Bahasa Indonesia). Guardians signed the informed consent on behalf of participants younger than 18 years old. MIT and the University of Washington provided feedback to ISWP by e-mail in December 2017. Overall, the MIT team felt the questions were useful but noted several items which were difficult for respondents to answer. For example, it was not easy for users to remember when they received their chair, but they could give a date range; users could not easily remember when they received their diagnosis but could estimate the number of years; and some questions were double-barreled (i.e., an item that asks two or more questions at the same time, and each one can be answered differently [52]). They also revised some questions to better meet their study objectives.

## Finalizing the questionnaires and launch

Feedback from the 2017 pilots was incorporated along with input from domain experts in instrument development at the University of Washington and University of Pittsburgh [51, 52]. The Evidence-based Practice Working Group provided additional feedback, resulting in final versions of the questionnaires prepared in March 2018 and presented to the ISWP

Evidence-based Practice Working Group on April 5, 2018 [51, 52]. The Excel workbooks and interviewer guides also were updated.

Table 3 presents a description of the launched MUD short form with 29 data points and standard version with 38 data points. The forms, fillable Excel workbooks, and questionnaire guides are available free of charge on ISWP's site: https://wheelchairnetwork.org/resource-library/mud/.

## ISWP Minimum Uniform Dataset preliminary translation

Forward translation of the MUD into French and Spanish was conducted. These translations were done in response to the request of two initiatives that were looking for a tool that they could quickly implement. First, a 2017 version of the short form is available in French. The translation was provided by Université de Montréal, CHU Sainte-Justine Centre de réadaptation Marie Enfant. It has not been forward-backward-forward translated, and there have been several changes to questions which are reflected in the 2018 short form. Second, UCP Wheels for Humanity continued to use the 2017 standard version (40 data points) in the full-study phase of its Wheelchair User's Voice Project. The questions were translated into Spanish and incorporated into a larger dataset which was administered in El Salvador in 2019. The short version of the questionnaire was translated into Spanish in 2019. Fig 3 presents the MUD's development, validation, and translation steps that were accomplished and the future work steps to be completed.

The English, Spanish, and French MUD versions are open-access resources that may be used by other groups for validation and/or to complete or conduct a full translation cycle [58].

## Discussion, limitations and future work

The International Society of Wheelchair Professionals developed a Minimum Uniform Dataset (MUD) for Wheelchair Service in English, Spanish, and French. This is the first measurement tool of its kind intended to promote global wheelchair service data collection and developed iteratively with input from global stakeholders (e.g., service providers in LMICs, researchers, service managers, users, NGOs, manufacturers). We believe that this collaboration will be key for a successful implementation; previous outcome measures research has identified the engagement of clinicians, as well as wheelchair users, managers, and providers, in the development of this type of clinical dataset as key for the success [18, 59]. The MUD may be used by ISWP members and allies to gather the short- and long-term evidence that is needed to provide guidance for policy and inform allocation of scarce resources into appropriate wheelchair provision, especially in LMICs [4, 12, 14]. The MUD has the potential to contribute to closing the *Policy Implementation Monitoring Gap* which has been defined as a lack of explicit disability disaggregated indicators for monitoring and evaluation at the national, regional, and global-levels [5, 14, 27, 49, 60]. The information could be used to accredit wheelchair services in a similar manner as how the Joint Commission and Commission on Accreditation of Rehabilitation Facilities International use the Uniform Dataset for Medical Rehabilitation in the US in their service accreditation processes [60]. Aggregated data also may be used to prove the conceptual framework that was developed to illustrate the factors that affect the cost effectiveness of wheelchair provision [61]. Last, the MUD may contribute to understanding at a system's level the demand and supply of wheelchair products and services that can inform procurement, innovation, and drive investment [29, 62]. As demonstrated in the UK, national and uniform wheelchair service data allow decisions to better align supply and demand with appropriate resource allocation [59].

**Table 3. ISWP MUD questionnaires (bold X indicates data point in standard version only).** This table has been reproduced from the RESNA Conference Abstract [53].

| Data Point | Short Form | Standard Version |
|---|---|---|
| **Client and wheelchair clinic information** | | |
| Client ID | X | X |
| Client country | X | X |
| Wheelchair clinic ID | X | X |
| Wheelchair clinic country | X | X |
| **Date completed** | X | X |
| **Purpose of Visit**\* | X | X |
| **Name of individual/organization referring client** | | **X** |
| **Demographics** | | |
| Age/approximate age | X | X |
| Gender\* | X | X |
| Education\* | X | X |
| Employment\* | X | X |
| Living situation\* | X | X |
| **Reasons for Assistance** | X | X |
| How long need something to help walk/move | X | X |
| Diagnosis\* | X | X |
| Year diagnosis received | X | X |
| Difficulty walking long distance (100 meters) | X | X |
| **Assistance when outdoors/inside**\* | | **X** |
| **Items used to help walk/move** (*list of 10 mobility aids. Presented as one question with 6 parts*) | | |
| Own item | X | X |
| Use indoors | X | X |
| Use outside | X | X |
| Use for more than one year | X | X |
| Number of days per week used | X | X |
| Number of hours per day used | X | X |
| **Questions for current wheelchair users** | | |
| Degree of difficulty pushing\* | X | X |
| How client typically pushes wheelchair\* | X | X |
| If not pushing with arms reasons why\* | X | X |
| Places where wheelchair is used\* | X | X |
| Distance traveled each day in wheelchair\* | | **X** |
| Whether public/private transportation is used\* | | **X** |
| If public/private transportation used, what kind\* | | **X** |
| Whether client has ever fallen out of wheelchair\* | | **X** |
| How person received wheelchair\* | X | X |
| Agreement statements regarding wheelchair\* | X | X |
| Level of satisfaction with wheelchair\* | X | X |
| Wheelchair training received\* | | **X** |
| Manufacturer name, model, serial number of item used to help person walk/move\* | | **X** |
| Manufacturer name, make, model name of cushion and/or postural support device (s) used\* | | **X** |

\*Close-ended question.

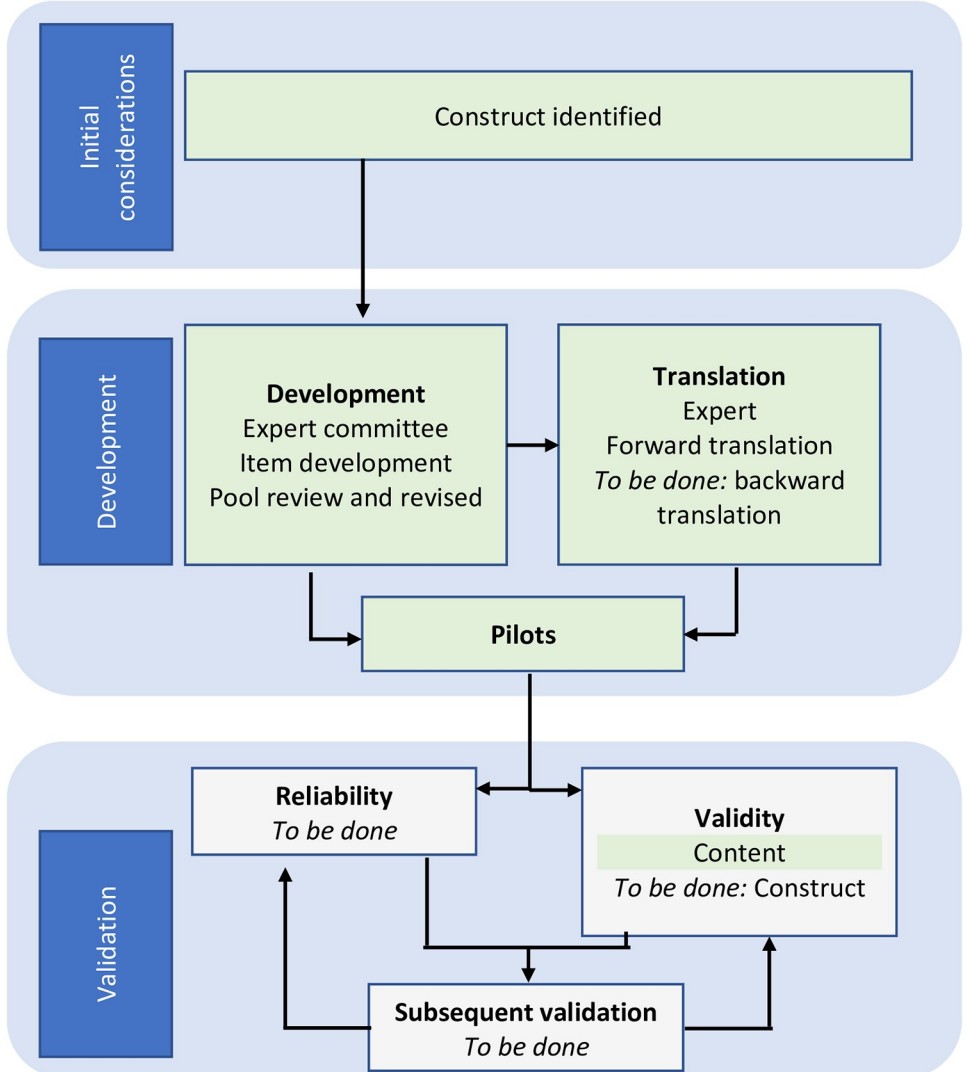

**Fig 3. ISWP's Minimum Uniform database accomplished and future development, validation, and translation steps.** Green shaded boxes indicate the completed steps. Adapted from [58].

The pilots conducted with the MUD in Kenya and Indonesia indicate that its use appears to be feasible. In addition, preliminary data from the registry in the US has demonstrated that administering the Wheeled Mobility Questionnaire is feasible in the clinical setting and does not add burden to the clinical routine [42]. The dataset being used in the US has 31 questions as compared to 29 questions in the short version MUD and 38 questions in the standard version MUD [42]. They differ in that the US dataset has context-specific questions [42], and the MUD is intended to be context-neutral. As expressed by ISWP members, there is an opportunity to develop the data collection and sharing system [53].

Experiences from large multicenter datasets have reported the need for sustained efforts to oversee the data collection quality to improve its reliability [17, 21, 63]. Currently, ISWP's MUD has accompanying written instructions on how to use it. Future work in data collector training, periodical centralizing of the global data, and frequent auditing is needed to

maximize data quality. Specifically, pending resource allocation, ISWP could conduct training for personnel who will administer the data collection tool [18] so it is used as intended [42]. The MUD presents organizations such as ISWP with the opportunity of a global data collection centralized repository. Instead of offering the opportunity to provide information when they want, ISWP could contact all MUD users quarterly or every 6 months to gather the data collected [59]. In addition, ISWP dedicated personnel could check for errors and missing data to request service providers/MUD users to correct as needed [18].

A potential limitation that this MUD presents is that it is a self-report measure, which is expected to be mitigated by having a large sample size [42] that will only result from a large uptake. However, future measurements are needed to ensure the reliability and validity of the MUD before it can serve as a reference on a global scale. The MUD's reliability is unknown, and the French and Spanish translations were done only forward (i.e. English to Spanish and French) [53]. Future work should include evaluating the psychometric properties of the MUD [53] and a complete forward-backward-forward translation process into other languages. As English is not the local language in most LMIC settings, translation of the MUD is important. By using a common MUD centralized at ISWP, the organization has the potential of gathering, analyzing, and reporting on non-English evidence. This is especially important as non-English evidence is commonly excluded from publications and analysis [13, 15].

Currently, ISWP members are encouraged to contribute to the MUD's psychometrics properties assessment and/or complete translation process. These steps must precede the use of the questionnaires in their practices. The tool is recommended for use at the assessment for a new device and regularly during follow-up and maintenance visits. The tool should be integrated into research protocols evaluating or comparing seating and mobility interventions.

ISWP could explore collaborations to complete the validation and translation steps in settings that have taken steady steps towards improving their wheelchair sector. For instance, Tajikistan, the Philippines, and Romania have conducted wheelchair sector situational analyses and indicated their need for reliable data [48, 64]. Last, ISWP could explore the feasibility of offering a service, in different languages, where member organizations could pay a nominal fee to conduct the follow-ups [42]. Future work should involve local service providers and grassroots organizations to ensure that the data collected is responding to the needs of local wheelchair providers and users [14, 25].

## Conclusion

A Minimum Uniform Dataset was developed and piloted by global ISWP stakeholders, and it is available in a short version with 29 data points and a standard form with 38 data points in English, Spanish, and French from www.wheelchairnetwork.org. Upon the validation and complete translation process of the MUD, and with appropriate resources, ISWP has the potential to manage the MUD as a global resource to bridge the gap in wheelchair service evidence that is needed to drive investment and foster the change that is needed so wheelchair users can exercise their human right to personal mobility in LMICs.

## Supporting information

**S1 File. ISWP member survey.**
(DOCX)

**S2 File. Minimum Uniform Dataset questionnaire in Bahasa.**
(DOCX)

**S3 File.**
(DOCX)

**S1 Dataset.**
(XLSX)

**S2 Dataset.**
(DOCX)

## Acknowledgments

The authors thank the ISWP Evidence-based Practice Working Group, Data Collection, and Comparative Effectiveness Research Subcommittee members for helping to develop the questionnaires: Kavi Bhalla, Johns Hopkins University; Johan Borg, Lund University; Nathan Bray, Centre for Health Economics and Medicine Evaluation; Molly Broderson, Free Wheelchair Mission; Mark Harniss, University of Washington; Kristi Haycock, Latter-day Saint Charities; Astrid Jenkinson, Motivation UK; Sara Múnera, Whee; Karen Reyes, World Health Organization; Chandra Whetstine, World Vision; and Eric Wunderlich, Latter-day Saint Charities. We thank LeTourneau University staff led by Karen Rispin, BethanyKids Relief and Rehabilitation, Joy Town School for Students with Disabilities, and the Wheelchair User's Voice Project, led by UCP Wheels for Humanity and implemented by CITE-MIT and the University of Pittsburgh, for piloting the questionnaires. We also thank University of Montreal representatives Paula Rushton and Geneviève Daoust for the French translation of the short form; Sara Munera, Whee, for the translation of the short form into Spanish; Dagmar Amtmann, University of Washington research professor experienced in developing patient-reported outcome measures, for reviewing the questionnaire; and University of Pittsburgh faculty and staff Deepan Kamaraj and Christina Zigler.

## Author Contributions

**Conceptualization:** Padmaja Kankipati, Patricia Karg, Karen Rispin, Richard M. Schein, Mary Goldberg, Jonathan Pearlman.

**Data curation:** Nancy Augustine.

**Formal analysis:** Nancy Augustine.

**Funding acquisition:** Jonathan Pearlman.

**Investigation:** María Luisa Toro-Hernández, Karen Rispin, Mary Goldberg.

**Methodology:** Nancy Augustine, Padmaja Kankipati, Patricia Karg, Karen Rispin, Richard M. Schein, Jonathan Pearlman.

**Project administration:** Nancy Augustine, Krithika Kandavel, Megan E. D'Innocenzo.

**Resources:** Mary Goldberg.

**Supervision:** Megan E. D'Innocenzo, Mary Goldberg.

**Writing – original draft:** María Luisa Toro-Hernández, Nancy Augustine, Krithika Kandavel.

**Writing – review & editing:** Padmaja Kankipati, Patricia Karg, Karen Rispin, Richard M. Schein, Megan E. D'Innocenzo, Mary Goldberg, Jonathan Pearlman.

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
