## [Decision Letter · Decision Letter 0]

30 Apr 2020

PONE-D-20-02442

Development of a Minimum Uniform Dataset applicable to the international wheelchair sector

PLOS ONE

Dear Dr Pearlman,

Thank you for submitting your manuscript to PLOS ONE. After careful consideration, we feel that it has merit but does not fully meet PLOS ONE’s publication criteria as it currently stands. Therefore, we invite you to submit a revised version of the manuscript that addresses the points raised during the review process.

Both reviewers have appreciated the overall study. Yet they are a number of major issues that need to be carefully addressed, both concerning the methodology and the presentation and discussion of the results. 

We would appreciate receiving your revised manuscript by Jun 14 2020 11:59PM. To enhance the reproducibility of your results, we recommend that if applicable you deposit your laboratory protocols in protocols.io, where a protocol can be assigned its own identifier (DOI) such that it can be cited independently in the future. For instructions see: http://journals.plos.org/plosone/s/submission-guidelines#loc-laboratory-protocols

We look forward to receiving your revised manuscript.

Kind regards,

Sara Rubinelli

Academic Editor

PLOS ONE

Journal Requirements:

2.  In your methods section, please describe how feedback on the questionnaire was collected; provide more information on questionnaire validation (for example on whether  internal consistency and reliability were assessed) ; and describe how the samples used for pilot testing was recruited and selected. "

3. Please provide additional details regarding participant consent. As your study included minors, state whether you obtained consent from parents or guardians.

4. Please include additional information regarding the survey or questionnaire used in the study and ensure that you have provided sufficient details that others could replicate the analyses. For instance, as you developed different questionnaires as part of this study (both the ISWP Member Survey and the  Minimum Uniform Dataset Questionnaire)  please include a copy as Supporting Information, both in English and in  Bahasa. Please also explain in more detail the methods used for its translation. "

6. Thank you for stating the following in the Competing Interests section:

'NO authors have competing interests' 

We note that one or more of the authors are employed by a commercial company: Specialized Mobility Operations and Innovation Pvt. Ltd.

Additional Editor Comments (if provided):

Reviewers' comments:

Reviewer's Responses to Questions

**Comments to the Author**

1. Is the manuscript technically sound, and do the data support the conclusions?

Reviewer #1: Yes

Reviewer #2: Partly

2. Has the statistical analysis been performed appropriately and rigorously? 

Reviewer #1: I Don't Know

Reviewer #2: N/A

3. Have the authors made all data underlying the findings in their manuscript fully available?

Reviewer #1: No

Reviewer #2: Yes

4. Is the manuscript presented in an intelligible fashion and written in standard English?

Reviewer #1: Yes

Reviewer #2: Yes

5. Review Comments to the Author

Reviewer #1: I am unfamiliar with statistical analysis on development of questionnaires and related restrictions on including datasets. Citations/literature on this would be helpful.

General comments:

The text contains a lot of jargon that people outside this area would be unfamiliar with – wheelchair sector, wheelchair stakeholders, etc. – that should be defined in the introduction. I am in the SCI field and I’ve never heard the term “wheelchair sector” and I’m unclear as to who a stakeholder is. Also, if “wheelchair sector” refers to individuals what use wheelchairs for mobility, it should be replaced with people-first language (“individuals that use wheelchairs” etc.).

The introduction could benefit from restructuring/reorganization: it’s long and there’s a lot of (necessary) information presented, but it’s hard to keep track of the aims of the paper and some of the ideas are repeated in each paragraph. For example, you talk about datasets in paragraphs 1 and 3, use of AP is introduced in paragraphs 1, 2, and 3, LMICs are discussed in paragraphs 2 and 3, and that there is no comparable research for clinicians to use is discussed in paragraphs 1 and 3. It would be easier to follow/understand if the structure were first paragraph – unmet need/significance; second paragraph: current datasets/evidence available; third paragraph: what is missing from datasets and why is it necessary.

Methods – there should be a sentence that summarizes this methodology in the beginning (needs assessment, rounds of pilot testing, refinement, etc.) with citations so the reader can know this methodology is standard for development and application of questionnaires/surveys.

Regarding the discussion – there was a lot of emphasis in the introduction on AP but no mention of it in the discussion. If it’s necessary and now included in the dataset, how is that going to improve the lives of individuals with SCI? How is that going to change rehabilitative outcomes?

Specific comments

31 Rehabilitation is a strategy to promote functioning and a prerequisite to achieve health and well

32 being for all (1)

“functioning” is awkward, and the tense doesn’t agree with the rest of the sentence – maybe “functional gains” or something similar is better

41 outcome measures which are useful to compare rehabilitation outcomes among type of services

“which are useful” should be replaced with “in order to”

46 hospitalization, diagnostic, functional status data, but do not measure assistive technology use

47 at discharge (7, 8, 10, 12). The use of assistive products (AP) has been identified globally as an

The abstract and line 46 have “assistive technology” but the abbreviation is for assistive products – should be consistent throughout

52 For many people, acc

ess to AP, and learning how to use them, is needed to improve functioning

Change “functioning”

67 rehabilitation services both in quantity and quality (4). A full picture of the wheelchair sector is

68 necessary for strategic planning and the capacity building to realize the right to personal mobility

“a full picture” of what? AP use/needs?

69 (15, 30). In fact, a global wheelchair sector stakeholders meeting in 2018 identified as one of the

70 sector’s 10 priorities to:

Who is in the sector and who is a stakeholder? What was the meeting?

74 to support current recommendations on wheelchair service delivery models, and the evidence

Recommendations by whom? What is a wheelchair service delivery model?

79 measurements and had high attrition rates, making it impossible to aggregate the data (36-39).

I don’t think I’ve ever seen attrition used in this manner and I’m unclear – turnover is high among individuals administering the assessment? Or are individuals that participate lost to follow-up?

97 To help bridge the gap in wheelchair provision data gathered and comparable in LMIC, the

98 International Society of Wheelchair Professionals (50) developed a Minimum Uniform Dataset

99 with the objective to promote the global use of a common language to strengthen the evidence

100 (29) and to foster comparison among interventions (33).

Something similar to this statement should be moved earlier in the introduction, there’s a lot of background presented and it’s very easy to lose sight of the message

101 present the development process and final composition of ISWP’s

I’m assuming this is an abbreviation for International Society of Wheelchair Professionals but it wasn’t defined on line 98

117 On May 5, 2015, ISWP sent an e-mail to 353 ISWP wheelchair sector stakeholders who are ISWP […]

121 identified, aggregated data; and suggested data to be collected. Forty-one individuals (41)

122 responded, for an 11.6% response rate.

Is this typical survey method and a typical response rate for this type of analysis? It seems low and maybe would have benefitted from multiple emails/methodologies/etc. How can we be sure this is a representative sample of clinicians/professionals necessary to make the standards valid?

137 satisfaction, feedback or impact surveys

Need comma after “feedback”

197 The 2016 pilot feedback and additional working group and subcommittee review and input

198 resulted in a standard version of the MUD with 40 data points and a short form with 28 data point

It’s unclear why a short form and standard form were considered necessary

233 and some questions were double barreled.

What is “double barreled”?

238 Feedback from the 2017 pilots was incorporated, along with input from domain experts in

239 instrument development at the University of Washington and University of Pittsburgh.

A citation of some sort would be useful here, to demonstrate this is the standard method for modifying survey questionnaires based on pilot data

271 tool of its kind intended to promote global wheelchair service data collection and developed

272 iteratively with input from global stakeholders

Again – it’s unclear who the stakeholders are, and what “wheelchair service” refers to

274 has been identified as key for the success (8, 56).

Are you saying it’s already been successful? Or the collaboration will make it successful?

279 disaggregated by disability (5, 22). In addition, as the global wheelchair sector strives to set

280 service standards (31) […]

287 to illustrate the factors that affect the cost effectiveness of wheelchair provision (58).

These lines are all related to the same idea (standards/development of best practices), it can be decreased to one sentence. It’s speculation

294 feasible. In addition, preliminary data from the registry in the US has shown that administering

“shown” should be replaced with a more quantitative word

296 the clinical routine (44). The dataset being used in the US has 31 questions as compared to 29

297 questions in the short version MUD and 38 questions in the standard version MUD (44).

Why is the dataset different?

303 has accompanying written instructions on how to use it. Future work is needed to maximize data

304 quality.

What specifically needs to be improved and why?

312 A potential limitation that this MUD presents is that it is a self-report measure, which is expected

It seems a limitation is also the small response during creation of the questionnaire by stakeholders, and the fact that implementation is mostly limited to ISWP stakeholders. How do the authors propose to get this dataset used by a wider audience?

332 grass roots organizations of people with disabilities who can contribute to gathering, monitoring

333 and evaluating data points in addition to providing users’ perspective (22, 27).

Is improving their own care and health outcomes really an appropriate burden to give individuals with SCI? It is beneficial to understand their input and use it to guide best practices, but it’s not the responsibility of marginalized/unempowered individuals to correct the system that disenfranchises them.

Reviewer #2: Thank you for the opportunity to review your manuscript. The paper entitled "Development of a Minimum Uniform Dataset applicable to the international wheelchair sector" reports essential findings concerning techniques for unstable intertrochanteric fractures. This study aims to provide a Minimum Uniform Dataset (MUD) for wheelchair services worldwide with the intent to gather data that is comparable globally. The paper is worth to be published in PONE after major revision. There is a global need to inform policy, practice and direct investments based on an evidence that might be gained through the present approach.

The translation process needs to be reconsidered as they might still bear a significant risk of bias in the current state. Afterwards, the analysis of the questionnaires reliability and validity needs to be adjusted and reported in the manuscript.

TITLE

-

INTRODUCTION

-

METHODS AND RESULTS

- Please provide a flowchart or table indicating the response rate of your Member survey for each item or a bar plot giving the absolute counts per item.

- Regarding the occupation, did you offer to select more than one option?

- Did you define criteria to exclude any response with a certain percentage of missing information? If yes, please specify.

- Please explain in detail why you decided not to forward-backward-forward translate the questionnaire before finalizing the translated versions.

- For a better understanding, please provide a detailed flowchart of the process including the present status and the steps missing before completing a state of the art questionnaire development process (please consider to refer to (https://www.who.int/substance_abuse/research_tools/translation/en/ and the chart attached for further information).

- Please provide measurements of the reliability (internal consistency, test-retest reliability and inter-rater reliability) and validity (content and construct validity) of the questionnaires in different languages.

DISCUSSION

- The study aims to develop a Minimum Uniform Dataset (MUD) for wheelchair services worldwide with the intent to gather data that is comparable globally.

- To ensure the comparability between the different translations, a forward-backward-forward translation process is recommended as the translation process might bias the results

- Further measurements are needed to ensure the reliability and validity of the questionnaire before the MUD is ready to serve as a reference on a global scale

CONCLUSION

- The present results and steps completed are promising, and the analysis is well designed.

- However, the questionnaires are difficult to compare due to missing further characteristics of the different language versions.

- Without providing further information, the questionnaires aims are challenging to fulfil, and the conclusions derived from the future data still bear a significant risk of bias.

- These limitations are partwise reflected in conclusion. Nevertheless, in the current state, the promotion of the MUD globally with the intention to inform policy, practice and direct investments appear challenging.

TABLES

-

APPENDICES

-

REFERENCES

-

I am looking forward to the revised manuscript.

6. PLOS authors have the option to publish the peer review history of their article (what does this mean?). If published, this will include your full peer review and any attached files.

Reviewer #1: No

Reviewer #2: No

---

## [Author Response · Author response to Decision Letter 0]

23 Jul 2020

July 19, 2020

Dear Reviewers and Editor:

We appreciate your thoughtful comments and recommendations. We have responded to them point-by-point and modified the manuscript using track changes. The line number reference in the response correspond to the numbering in the market version. We also uploaded an unmarked version of the revised manuscript. 

Thank you.

Response to Reviewer’s 1 comments:

Reviewer #1: 

Comment 1. I am unfamiliar with statistical analysis on development of questionnaires and related restrictions on including datasets. Citations/literature on this would be helpful.

Response 1. Two references have guided our work and are now cited in the methods section: 

Streiner DL, Norman GR, Cairney J. Selecting items. In: Streiner DL, Norman GR, Cairney J, editors. Health Measurement Scales: A practical guide to their development and use. 5th ed. New York: Oxford University Press; 2015.

Portney LG, Watkins MP. Surveys and Questionnaires. In: Portney LG, Watkins MP, editors. Foundations of Clinical Research Applications to Practice. Third ed. New Jersey: Pearson Prentice Hall; 2008.

General comments:

Comment 2. The text contains a lot of jargon that people outside this area would be unfamiliar with – wheelchair sector, wheelchair stakeholders, etc. – that should be defined in the introduction. I am in the SCI field and I’ve never heard the term “wheelchair sector” and I’m unclear as to who a stakeholder is. Also, if “wheelchair sector” refers to individuals what use wheelchairs for mobility, it should be replaced with people-first language (“individuals that use wheelchairs” etc.).

Response 2. The wheelchair sector and stakeholders are now defined in the introduction.

Lines 144-154: “A full picture of the need for appropriate wheelchair products and services, including the policies and personnel that support them, is necessary for strategic planning and capacity building to realize the human right to personal mobility (7, 48)”

Lines 154-159: “In fact, a global wheelchair sector stakeholders (e.g., users, WHO, aid agencies, academia, international and local NGOs, humanitarian organizations, professional organizations, service providers, governments) meeting in 2018 held to reflect on past achievements, challenges, current initiatives, and to strategize a future of better access to wheelchairs, identified one of the global priorities to be:” 

Comment 3. The introduction could benefit from restructuring/reorganization: it’s long and there’s a lot of (necessary) information presented, but it’s hard to keep track of the aims of the paper and some of the ideas are repeated in each paragraph. For example, you talk about datasets in paragraphs 1 and 3, use of AP is introduced in paragraphs 1, 2, and 3, LMICs are discussed in paragraphs 2 and 3, and that there is no comparable research for clinicians to use is discussed in paragraphs 1 and 3. It would be easier to follow/understand if the structure were first paragraph – unmet need/significance; second paragraph: current datasets/evidence available; third paragraph: what is missing from datasets and why is it necessary.

Response 3. We have followed your recommendation and the introduction has been revised for clarity and conciseness.

Comment 4. Methods – there should be a sentence that summarizes this methodology in the beginning (needs assessment, rounds of pilot testing, refinement, etc.) with citations so the reader can know this methodology is standard for development and application of questionnaires/surveys.

Response 4. This information is now included in Lines 175-177: “The iterative process included initial question development, pilots, revisions based on pilots’ results, and launch (Figure 1) (51, 52).”

Comment 5. Regarding the discussion – there was a lot of emphasis in the introduction on AP but no mention of it in the discussion. If it’s necessary and now included in the dataset, how is that going to improve the lives of individuals with SCI? How is that going to change rehabilitative outcomes?

Response 5. We believe that we have addressed this in the first paragraph of the introduction but may have not been explicit enough. We now included “wheelchair services” and/or “wheelchair products” in lines 147, 442, 479.

Specific comments

Comment 6. 31 Rehabilitation is a strategy to promote functioning and a prerequisite to achieve health and well 32 being for all (1) “functioning” is awkward, and the tense doesn’t agree with the rest of the sentence – maybe “functional gains” or something similar is better

Response 6. We consider that using the term “functioning” is most appropriate to the current WHO Rehabilitation 2030 initiative definition which is based on the ICF. The sentence was revised to “The World Health Organization (WHO) defines rehabilitation as a strategy to optimize an individual’s functioning (1). WHO has also recognized that rehabilitation is a prerequisite to achieve health and well-being for all (1).”

Comment 7. 41 outcome measures which are useful to compare rehabilitation outcomes among type of services “which are useful” should be replaced with “in order to”

Response 7. Change included.

Comment 8. 46 hospitalization, diagnostic, functional status data, but do not measure assistive technology use 47 at discharge (7, 8, 10, 12). The use of assistive products (AP) has been identified globally as an The abstract and line 46 have “assistive technology” but the abbreviation is for assistive products – should be consistent throughout

Response 8. The text now uses assistive products consistently.

Comment 9. 52 For many people, access to AP, and learning how to use them, is needed to improve functioning Change “functioning”

Response 9. This information is now in Lines 43-54 and reads: “For many people, access to assistive products (AP), and learning how to use them, is needed to realize the right to fully participate in their communities (1, 7-9).” 

Comment 10. 67 rehabilitation services both in quantity and quality (4). A full picture of the wheelchair sector is 68 necessary for strategic planning and the capacity building to realize the right to personal mobility “a full picture” of what? AP use/needs?

Response 10. A description of the wheelchair sector is now included in Lines 144-154: “A full picture of the wheelchair sector (i.e. met and unmet need, products, service provision, personnel, and policies) is necessary for strategic planning and the capacity building to realize the right to personal mobility (7, 48).”

Comment 11. 69 (15, 30). In fact, a global wheelchair sector stakeholders meeting in 2018 identified as one of the 70 sector’s 10 priorities to: Who is in the sector and who is a stakeholder? What was the meeting?

Response 11. The wheelchair sector is now described as mentioned in the previous comment. Additional information on the goal of the meeting and examples of the stakeholders is now presented (Lines 155-159): “In fact, a global wheelchair sector stakeholders (e.g., users, WHO, aid agencies, academia, international and local NGOs, humanitarian organizations, professional organizations, service providers, governments) meeting in 2018 held to reflect on past achievements, challenges, current initiatives, and to strategize a future of better access to wheelchairs, identified one of the global priorities to be:

Comment 12. 74 to support current recommendations on wheelchair service delivery models, and the evidence. Recommendations by whom? What is a wheelchair service delivery model?

Response 12. Lines 78-124 have been revised to: “There is little research to support current wheelchair service delivery recommendations (e.g. WHO guidelines, professional organization guidelines), and the evidence that exists commonly involves small sample sizes, inconsistent definitions, and different outcome measurements (30, 31). ”

Comment 13. 79 measurements and had high attrition rates, making it impossible to aggregate the data (36-39). I don’t think I’ve ever seen attrition used in this manner and I’m unclear – turnover is high among individuals administering the assessment? Or are individuals that participate lost to follow-up?

Response 13. Lines 124-127 now read: “Efforts targeted at LMICs to measure wheelchair provision outcomes longitudinally have used different measurements and a significant group of people were lost to follow-up, making it impossible to aggregate the data (34-37).”

Comment 14. 97 To help bridge the gap in wheelchair provision data gathered and comparable in LMIC, the 98 International Society of Wheelchair Professionals (50) developed a Minimum Uniform Dataset 99 with the objective to promote the global use of a common language to strengthen the evidence 100 (29) and to foster comparison among interventions (33). Something similar to this statement should be moved earlier in the introduction, there’s a lot of background presented and it’s very easy to lose sight of the message

Response 14. We included in Lines 57-58: “Specific to wheelchairs, there is a need for comparable wheelchair provision data, mostly in LMICs, to strengthen the evidence on different interventions (15, 16).”

Comment 15. 101 present the development process and final composition of ISWP’s

I’m assuming this is an abbreviation for International Society of Wheelchair Professionals but it wasn’t defined on line 98 117 On May 5, 2015, ISWP sent an e-mail to 353 ISWP wheelchair sector stakeholders who are ISWP […] 121 identified, aggregated data; and suggested data to be collected. Forty-one individuals (41)

122 responded, for an 11.6% response rate.

Is this typical survey method and a typical response rate for this type of analysis? It seems low and maybe would have benefitted from multiple emails/methodologies/etc. How can we be sure this is a representative sample of clinicians/professionals necessary to make the standards valid?

Response 16. ISWP’s acronym is mentioned first in the introduction in line 1604 We revised lines 267-270 to clarify that the respondents represented all stakeholder groups and a broad variety of countries.

Lines 267-270: “Forty-one individuals (41) responded (11.6%) representing all stakeholder groups and 18 countries, providing a representative sample for this first step of identifying the types and extent of data collection.” 

Lines 279-280: “The survey was distributed only in English; this may have prevented stakeholders whose first language is not English from responding.”

Comment 16. 137 satisfaction, feedback or impact surveys Need comma after “feedback”

Response 16. Comma added.

Comment 17. 197 The 2016 pilot feedback and additional working group and subcommittee review and input 198 resulted in a standard version of the MUD with 40 data points and a short form with 28 data point It’s unclear why a short form and standard form were considered necessary 233 and some questions were double barreled.

What is “double barreled”?

Response 17. The rationale behind creating two version is now presented in lines 366-368: “The subcommittee decided to offer a standard and short form version to provide flexibility while promoting consistency in the implementation as time constraints in services may vary across LMICs.” 

Double-barreled is now defined in line 405: “some questions were double-barreled (i.e. an item that asks two or more questions at the same time and each one can be answered differently(52)).”

Comment 18. 238 Feedback from the 2017 pilots was incorporated, along with input from domain experts in 239 instrument development at the University of Washington and University of Pittsburgh. A citation of some sort would be useful here, to demonstrate this is the standard method for modifying survey questionnaires based on pilot data.

Response 18. Streiner et a. 2015 and Portney and Watkins 2008 are referenced now in this section. 

Comment 19. 271 tool of its kind intended to promote global wheelchair service data collection and developed 272 iteratively with input from global stakeholders

Again – it’s unclear who the stakeholders are, and what “wheelchair service” refers to.

Response 19. Line 454-457 was revised to: “This is the first measurement tool of its kind intended to promote global wheelchair service data collection and developed iteratively with input from global stakeholders (e.g. service providers in LMICs, researchers, service managers, users, non-government organizations, manufacturers).”

Comment 20. 274 has been identified as key for the success (8, 56).

Are you saying it’s already been successful? Or the collaboration will make it successful?

Response 20. Lines 457-460 now reads: “We believe that this collaboration will be key for a successful implementation; previous outcome measures research has identified the engagement of clinicians, as well as wheelchair users, managers, and providers, in the development of this type of clinical dataset as key for the success (18, 58).”

Comment 21. 279 disaggregated by disability (5, 22). In addition, as the global wheelchair sector strives to set 280 service standards (31) […] 287 to illustrate the factors that affect the cost effectiveness of wheelchair provision (58). These lines are all related to the same idea (standards/development of best practices), it can be decreased to one sentence. It’s speculation.

Response 21. The information has been condensed in Lines 463-466: “The MUD has the potential to contribute to closing the Policy Implementation Monitoring Gap which has been defined as a lack of explicit disability disaggregated indicators for monitoring and evaluation at the national, regional, and global-levels (5, 14, 27)(49)(59). ”

Comment 22. 294 feasible. In addition, preliminary data from the registry in the US has shown that administering “shown” should be replaced with a more quantitative word

Response 22. Shown was changed to “demonstrated”

Comment 23. 296 the clinical routine (44). The dataset being used in the US has 31 questions as compared to 29 297 questions in the short version MUD and 38 questions in the standard version MUD (44). Why is the dataset different? 

Response 23. The following sentence was added in line 510 “They differ in that the US dataset has context-specific questions (42) and the MUD is intended to be context-neutral”

Comment 24. 303 has accompanying written instructions on how to use it. Future work is needed to maximize data 304 quality. What specifically needs to be improved and why?

Response 24. The sentence was expanded and the connector modified for clarity. Line 516-518 now reads “Future work in data collector training, periodical centralizing of the global data, and frequent audit is needed to maximize data quality. Specifically, …”

Comment 25. 312 A potential limitation that this MUD presents is that it is a self-report measure, which is expected. It seems a limitation is also the small response during creation of the questionnaire by stakeholders, and the fact that implementation is mostly limited to ISWP stakeholders. How do the authors propose to get this dataset used by a wider audience?

Response 25. We feel the number of responses to the initial survey to assess the types of data being collected and potential data types needed was adequate due to diverse stakeholder and geographical representation. The dataset was developed from additional steps that involved expert input, consensus and pilot testing. Implementation is planned as future work and addressed in the last paragraph of the discussion. 

Comment 26. 332 grass roots organizations of people with disabilities who can contribute to gathering, monitoring 333 and evaluating data points in addition to providing users’ perspective (22, 27). Is improving their own care and health outcomes really an appropriate burden to give individuals with SCI? It is beneficial to understand their input and use it to guide best practices, but it’s not the responsibility of marginalized/unempowered individuals to correct the system that disenfranchises them.

Response 26. The sentence was reworded: “Future work should involve local service providers and grass roots organizations to ensure that the data collected is responding to the needs of local wheelchair providers and users (14, 25).”

Response to Reviewer’s 2 comments:

Comment 27. Thank you for the opportunity to review your manuscript. The paper entitled "Development of a Minimum Uniform Dataset applicable to the international wheelchair sector" reports essential findings concerning techniques for unstable intertrochanteric fractures. This study aims to provide a Minimum Uniform Dataset (MUD) for wheelchair services worldwide with the intent to gather data that is comparable globally. The paper is worth to be published in PONE after major revision. There is a global need to inform policy, practice and direct investments based on an evidence that might be gained through the present approach. The translation process needs to be reconsidered as they might still bear a significant risk of bias in the current state. Afterwards, the analysis of the questionnaires reliability and validity needs to be adjusted and reported in the manuscript.

Response 27. To clarify that the translated versions are in a preliminary stage, the subheading “ISWP Minimum Uniform Dataset Preliminary Translation” was added in Line 396. 

The need to test psychometric properties of the questionnaire and complete translation steps is stated as future work in lines 533.

Comment 28. Please provide a flowchart or table indicating the response rate of your Member survey for each item or a bar plot giving the absolute counts per item.

Response 28. We added the following information in Lines 291-292: “Overall, the 13 questions had a response rate of 76% or higher. Two questions were responded by 76% of the respondents, one by 88%, and 10 questions were responded by 90% or higher“

Comment 29. Regarding the occupation, did you offer to select more than one option?

Response 29. No, it was a single-answer multiple-choice question: “select the occupation that best describes you”. Line 272 was revised to reflect this and now reads: Among 39 respondents who selected the occupation that best described them, 28% (n=11) were clinicians; 15% (n=6) worked for non-government organizations; 15% (n=6) were researchers; and 10% (n=4) were manufacturers.

Comment 30. Did you define criteria to exclude any response with a certain percentage of missing information? If yes, please specify.

Response 30. Line 301 now reads: “No criteria were set a-priori to delete items based on missing responses.”

Comment 31. For a better understanding, please provide a detailed flowchart of the process including the present status and the steps missing before completing a state of the art questionnaire development process (please consider to refer to (https://www.who.int/substance_abuse/research_tools/translation/en/ and the chart attached for further information).

Response 31. We added figure 3 in the new section ISWP Minimum Uniform Dataset Preliminary Translation. 

Comment 32. Please explain in detail why you decided not to forward-backward-forward translate the questionnaire before finalizing the translated versions. Please provide measurements of the reliability (internal consistency, test-retest reliability and inter-rater reliability) and validity (content and construct validity) of the questionnaires in different languages. The study aims to develop a Minimum Uniform Dataset (MUD) for wheelchair services worldwide with the intent to gather data that is comparable globally. To ensure the comparability between the different translations, a forward-backward-forward translation process is recommended as the translation process might bias the results. Further measurements are needed to ensure the reliability and validity of the questionnaire before the MUD is ready to serve as a reference on a global scale. The present results and steps completed are promising, and the analysis is well designed. However, the questionnaires are difficult to compare due to missing further characteristics of the different language versions. Without providing further information, the questionnaires aims are challenging to fulfil, and the conclusions derived from the future data still bear a significant risk of bias. These limitations are partwise reflected in conclusion. Nevertheless, in the current state, the promotion of the MUD globally with the intention to inform policy, practice and direct investments appear challenging.

Response 32. The title has been revised to “Preliminary steps of the development of a Minimum Uniform Dataset applicable to the international wheelchair sector”.

Edits have been made to clarify the steps completed and remaining, including the addition of the flowchart (figure 3) and that the completion of the translation and validation in the future work. We acknowledge the limitations of not having completed the psychometrics analysis and the complete translation cycle. Due to the open-access and global scope of the MUD, we see that there is value in publishing this preliminary translation steps, other groups may be able to build upon this work. This is now included in lines 448 “The English, Spanish, and French MUD versions are open-access resources that may be used by other groups for validation and/or to complete or conduct a full translation cycle (58).”

Line 530 was added: “However, future measurements are needed to ensure the reliability and validity of the MUD before it can serve as a reference on a global scale.”

Line 533 was expanded: “Future work should include evaluating the psychometric properties of the MUD (53) and a complete forward-backward-forward translation process into other languages.”

The last two paragraphs in the discussion section and the conclusion have been revised to reflect the fact that validation and translation are work in progress. 

Lines 540-542: “Currently, ISWP members are encouraged to contribute to the MUD’s psychometrics properties assessment and/or complete translation process. These steps must precede the use of the questionnaires in their practices.”

Line 545 “ISWP could explore collaborations to complete the validation and translation steps in settings that have taken steady steps towards improving their wheelchair sector.”

Line 577: “Upon the validation and complete translation process of the MUD, and with appropriate resources, ISWP has the potential to manage the MUD as a global resource to bridge the gap in wheelchair service evidence that is needed to drive investment and foster the change that is needed so wheelchair users can exercise their human right to personal mobility in LMICs.”

Journal Requirements:

Comment 33. Please ensure that your manuscript meets PLOS ONE's style requirements, including those for file naming. The PLOS ONE style templates can be found at https://journals.plos.org/plosone/s/file?id=wjVg/PLOSOne_formatting_sample_main_body.pdf and https://journals.plos.org/plosone/s/file?id=ba62/PLOSOne_formatting_sample_title_authors_affiliations.pdf

Response 33. Our manuscript files meet PLOS ONE’s style.

Comment 34. In your methods section, please describe how feedback on the questionnaire was collected; provide more information on questionnaire validation (for example on whether internal consistency and reliability were assessed) ; and describe how the samples used for pilot testing was recruited and selected. "

Response 34.

The methods now describe that feedback was consolidated by MUD administrators in Excel files based on regular calls with working group members who provided regular input on the dataset. Lines 372-378 state: “MIT and the University of Washington provided feedback to ISWP by e-mail in December 2017.”

Comments related to the MUD’s validation were addressed in responses to reviewer’s 2 comments #27, 31, 32

Comment 35. Please provide additional details regarding participant consent. As your study included minors, state whether you obtained consent from parents or guardians.

Response 35. Lines 346-347: Students from two schools who used a wheelchair were invited to participate in the pilot.

Lines 350-525 now read: “Participants 18 years or older signed the informed consent and participants younger than 18 provided assent and their guardian signed the written consent.”

Lines 384 now read: “Ethical approval and recruitment and consent processes were as described above.”

Lines 398-399 were added: “Guardians signed the informed consent on behalf of participants younger than 18 years old.”

Comment 36. Please include additional information regarding the survey or questionnaire used in the study and ensure that you have provided sufficient details that others could replicate the analyses. For instance, as you developed different questionnaires as part of this study (both the ISWP Member Survey and the Minimum Uniform Dataset Questionnaire) please include a copy as Supporting Information, both in English and in Bahasa. Please also explain in more detail the methods used for its translation. "

Response 36. Line 577 includes the website <www.wheelchairnetwork.org> where the current MUD versions are available.

Lines 397-399 now read: “The questionnaire was forward translated and administered in Bahasa, and written consent was obtained from all research participants.”

The following Supplemental Information has been included and captions referenced added in the text:

Supplemental information 1 ISWP Member Survey; referenced in line 264

Supplemental information 2 ISWP Member Survey de-identified information, referenced in line 270

Supplemental information 3 data collector MUD feedback from Kenya pilots in 2016 and 2017; referenced in line 361 and 391

Supplemental information 4 MUD in Bahasa; reference in line 400

Comment 37. We note that you have indicated that data from this study are available upon request. PLOS only allows data to be available upon request if there are legal or ethical restrictions on sharing data publicly. For information on unacceptable data access restrictions, please see http://journals.plos.org/plosone/s/data-availability#loc-unacceptable-data-access-restrictions.

Response 37. The development of the MUD did not involve the collection or analysis of data to support research findings, thus there is no data set to share or archive. 

Comment 38. Thank you for stating the following in the Competing Interests section:

'NO authors have competing interests' 

We note that one or more of the authors are employed by a commercial company: Specialized Mobility Operations and Innovation Pvt. Ltd.

Response 38. 

Revised Funding Disclosure Statement

This research was made possible by the generous support of the International Society of Wheelchair Professionals (ISWP) and the University of Pittsburgh. University of Pittsburgh scientists are working with the U.S. Agency for International Development (USAID) under sub-awards to develop the International Society of Wheelchair Professionals, a global network to ensure a level of standardization, certification and oversight, to teach and professionalize wheelchair services, and to build affiliations to put better equipment in the right hands. Since 2002, USAID has granted more than $45 million to improve wheelchairs and wheelchair services worldwide. The sub-awards are: Agreement No. APC-GM-0068 and Agreement No. APC-GM-0107, presented by Advancing Partners & Communities, a cooperative agreement funded through USAID under Agreement No. AIDOAA-A-12-00047, beginning Oct. 1, 2012; and FY19-A01-6024, presented through University Research Co. LLC Health Evaluation and Applied Research Development (HEARD) Project. HEARD is funded by United States Agency for International Development (USAID) under cooperative agreement number AID-OAA-A-17-00002. The project team includes prime recipient, University Research Co., LLC (URC) and sub-recipient organizations. The contents of the findings of this study are the sole responsibility of Advancing Partners & Communities, University Research Co., LLC and the University of Pittsburgh and do not necessarily reflect the view of USAID or the United States Government. Jonathan Pearlman (JP) is the recipient of the above-mentioned awards. 

The funder provided support in the form of salaries for authors NA, JP, MG, KK, MLTH, RS, MD, but did not have any additional role in the study design, data collection and analysis, decision to publish, or preparation of the manuscript. The specific roles of these authors are articulated in the ‘author contributions’ section. PK participated as a volunteer and not as part of her role with SMOI. 

Revised competing interest statement

NO authors have competing interests. PK affiliation with SMOI does not alter our adherence to PLOS ONE policies on sharing data and materials.

Comment 39. Please include captions for your Supporting Information files at the end of your manuscript, and update any in-text citations to match accordingly. Please see our Supporting Information guidelines for more information: http://journals.plos.org/plosone/s/supporting-information.

Response 39. Lines 790 onwards now includes the list of the Supporting information files

Comment 40. While revising your submission, please upload your figure files to the Preflight Analysis and Conversion Engine (PACE) digital diagnostic tool, https://pacev2.apexcovantage.com/. PACE helps ensure that figures meet PLOS requirements. To use PACE, you must first register as a user. Registration is free. Then, login and navigate to the UPLOAD tab, where you will find detailed instructions on how to use the tool. If you encounter any issues or have any questions when using PACE, please email us at figures@plos.org. Please note that Supporting Information files do not need this step.

Response 40. Figures 1-3 were inspected using PACE.

---

## [Decision Letter · Decision Letter 1]

26 Aug 2020

Preliminary steps of the development of a Minimum Uniform Dataset applicable to the international wheelchair sector

PONE-D-20-02442R1

Dear Dr. Pearlman,

We’re pleased to inform you that your manuscript has been judged scientifically suitable for publication and will be formally accepted for publication once it meets all outstanding technical requirements.

Kind regards,

Sara Rubinelli

Academic Editor

PLOS ONE

Additional Editor Comments (optional):

Reviewers' comments:

Reviewer's Responses to Questions

**Comments to the Author**

1. If the authors have adequately addressed your comments raised in a previous round of review and you feel that this manuscript is now acceptable for publication, you may indicate that here to bypass the “Comments to the Author” section, enter your conflict of interest statement in the “Confidential to Editor” section, and submit your "Accept" recommendation.

Reviewer #1: All comments have been addressed

Reviewer #2: All comments have been addressed

2. Is the manuscript technically sound, and do the data support the conclusions?

Reviewer #1: Yes

Reviewer #2: Yes

3. Has the statistical analysis been performed appropriately and rigorously? 

Reviewer #1: Yes

Reviewer #2: N/A

4. Have the authors made all data underlying the findings in their manuscript fully available?

Reviewer #1: Yes

Reviewer #2: Yes

5. Is the manuscript presented in an intelligible fashion and written in standard English?

Reviewer #1: Yes

Reviewer #2: Yes

6. Review Comments to the Author

Reviewer #1: (No Response)

Reviewer #2: Thank you for the opportunity to review the revised version your manuscript. The paper reports essential findings and provides a compelling basis for further studies and is worth to be published in PLOS One. The paper is appropriately designed and feasible. Some minor typo errors need the authors attention.

7. PLOS authors have the option to publish the peer review history of their article (what does this mean?). If published, this will include your full peer review and any attached files.

Reviewer #1: No

Reviewer #2: No

---

## [Editor Report · Acceptance letter]

3 Sep 2020

PONE-D-20-02442R1 

Preliminary steps of the development of a Minimum Uniform Dataset applicable to the international wheelchair sector 

Dear Dr. Pearlman:

I'm pleased to inform you that your manuscript has been deemed suitable for publication in PLOS ONE. Congratulations! Your manuscript is now with our production department. 

Kind regards, 

on behalf of

Dr. Sara Rubinelli 

Academic Editor

PLOS ONE